# Asymmetrical diversification of the receptor-ligand interaction controlling self-incompatibility in Arabidopsis

Maxime Chantreau[1], Céline Poux[1], Marc F Lensink[2], Guillaume Brysbaert[2], Xavier Vekemans[1], Vincent Castric[1]*

[1]CNRS, Univ. Lille, UMR 8198—Evo-Eco-Paléo, F-59000, Lille, France; [2]Univ. Lille, CNRS, UMR 8576 - UGSF - Unité de Glycobiologie Structurale et Fonctionnelle, F-59000, Lille, France

**Abstract** How two-component genetic systems accumulate evolutionary novelty and diversify in the course of evolution is a fundamental problem in evolutionary systems biology. In the Brassicaceae, self-incompatibility (SI) is a spectacular example of a diversified allelic series in which numerous highly diverged receptor-ligand combinations are segregating in natural populations. However, the evolutionary mechanisms by which new SI specificities arise have remained elusive. Using in planta ancestral protein reconstruction, we demonstrate that two allelic variants segregating as distinct receptor-ligand combinations diverged through an asymmetrical process whereby one variant has retained the same recognition specificity as their (now extinct) putative ancestor, while the other has functionally diverged and now represents a novel specificity no longer recognized by the ancestor. Examination of the structural determinants of the shift in binding specificity suggests that qualitative rather than quantitative changes of the interaction are an important source of evolutionary novelty in this highly diversified receptor-ligand system.

**\*For correspondence:**
vincent.castric@univ-lille.fr

**Competing interests:** The authors declare that no competing interests exist.

## Introduction

A central goal of biology is to understand the evolutionary forces and molecular processes by which new traits and functions emerge in living organisms. A major challenge toward this goal is that many cellular processes rely on interactions between molecular partners (protein-protein interactions or regulatory interactions between for example transcription factors and their binding sites; *Boyle et al., 2017*; *Courtier-Orgogozo et al., 2019*) rather than on the action of individual components in isolation. At present, the overall structure of protein-protein or regulatory interaction networks is just starting to be understood at the genome level in a handful of model organisms, and a general understanding of how the diversification of these networks led to the emergence of new biological functions in distinct lineages is crucially lacking (*Nooren and Thornton, 2003*; *Andreani and Guerois, 2014*). In two-component genetic systems, specificity of the interaction can be very tight such as for example in receptor-ligand interactions (*Laub and Goulian, 2007*) or bacterial toxin-antitoxin systems (*Aakre et al., 2015*). In these systems, the emergence of novel traits involves evolutionary modification of the interacting partners, leading to potential disruption of the interaction or to detrimental cross-talk, at least transiently (*Plach et al., 2017*). A central aspect of the diversification process therefore concerns the functional nature of the evolutionary intermediate. Two main scenarios have been proposed (*Aakre et al., 2015*). The first scenario posits that a functional change results from introduction of a mutation that first modifies one of the two partners, initially disrupting the functional interaction and leading to a non-functional intermediate. If fitness of this non-functional intermediate is not impaired too drastically, it may remain in the population until a compensatory mutation hits the second partner and rescues the interaction in a modified state,

resulting in the novel function. An alternative scenario though, is that the first mutation may broaden rather than inactivate specificity of one of the two interacting partners, transiently releasing constraint on the other before specificity of the first partner becomes restricted again, thus maintaining functionality of the system all along the process. Distinguishing between these two scenarios requires functional characterization of the ancestral evolutionary intermediates, which has been achieved in a very limited number of biological cases such as bacterial toxin-antitoxin systems (*Aakre et al., 2015*), acquisition of cortisol specificity of the mammalian glucocorticoid receptor (*Bridgham et al., 2009*) or mammalian retinoic acid receptors (*Gutierrez-Mazariegos et al., 2016*). The scenario of a promiscuous intermediate seems to have received better support, but whether this is a truly general pattern remains to be determined. A major limitation is that detailed population genetic models taking into account how natural selection is acting on variants of the two genes given their specific biological function are lacking for most genetic systems (*Aakre et al., 2015*; *Aharoni et al., 2005*; *Bloom and Arnold, 2009*; *Bridgham et al., 2009*; *Matsumura and Ellington, 2001*; *Matton et al., 2000*; *Sayou et al., 2014*).

Self-incompatibility (SI) in flowering plants is a prime biological system to investigate how functional diversification can proceed between two molecular partners over the course of evolution. SI has evolved as a strategy to prevent selfing and enforce outcrossing, thereby avoiding the deleterious effects of inbreeding depression (*Nettancourt, 1977*; *Kitashiba and Nasrallah, 2014*). In *Brassicaceae*, SI is genetically controlled by a single multiallelic non-recombining locus called the S-locus (*Nettancourt, 1977*). This locus encodes two highly polymorphic self-recognition determinant proteins: the female cell-surface 'receptor' S-LOCUS RECEPTOR KINASE (SRK) is expressed in stigmatic papillae cells (*Takasaki et al., 2000*) and the male 'ligand' S-LOCUS CYSTEIN RICH PROTEIN (SCR) is displayed at the pollen surface (*Schopfer et al., 1999*; *Takasaki et al., 2000*; *Takayama et al., 2000*). The SI response consists in pollen rejection by the pistil and is induced by allele-specific interaction between the SRK and SCR proteins when encoded by the same haplotype (S-haplotype, *Kachroo et al., 2001*; *Nasrallah and Nasrallah, 1993*; *Takayama et al., 2001*). These two interacting proteins exhibit a high degree of sequence variability and a large number of these highly diverged allelic variants segregate in natural populations of SI species (several dozens to nearly two hundred, *Busch et al., 2014*; *Castric and Vekemans, 2004*; *Lawrence, 2000*). This large allelic diversity of receptor-ligand combinations must have arisen through repeated diversification events, but the molecular process and evolutionary scenario involved in the diversification of an ancestral haplotype into distinct descendant specificities are still unresolved.

Three evolutionary scenarios for emergence of new self-incompatibility specificities have been proposed so far (*Charlesworth et al., 2005*). The 'compensatory mutation' scenario posits that diversification proceeds through self-compatible intermediates that is following transient disruption of the receptor-ligand interaction (*Figure 1—figure supplement 1A*; *Gervais et al., 2011*; *Uyenoyama et al., 2001*). Population genetic analysis confirmed that S-allele diversification is indeed possible through this scenario, but only under some combinations of model parameters (high inbreeding depression, high rate of self-pollination and low number of co-segregating S-alleles, *Gervais et al., 2011*). Under these restrictive conditions, the ancestral recognition specificity can be maintained in the long run along with the derived specificity, effectively resulting in allelic diversification. A second scenario was presented by *Chookajorn et al. (2004)*, who proposed that diversification results from the progressive reinforcement of SRK/SCR recognition capacity among slight functional variants of *SCR* and *SRK* that might segregate within the population. At the limit of the very low level of intra-allelic polymorphism observed in natural populations (*Castric et al., 2010*) this model bears similarity to the model of *Gervais et al. (2011)*, who also predicted the maintenance of an ancestral recognition specificity unchanged and the evolution of a new divergent specificity, except it assumes no SC intermediate. However, the progressive reinforcement process proposed by this model may rather result in the production of two descendant allelic specificities that are both functionally distinct from their ancestor (*Figure 1—figure supplement 1B*). A third scenario was proposed by *Matton et al. (1999)* and involves a dual-specificity intermediate (*Figure 1—figure supplement 1C*). This scenario was criticized on population genetics grounds because such a dual-specificity haplotype would recognize and reject more potential mates for reproduction than its progenitor haplotype resulting in lower reproductive success, and should therefore be disfavoured by natural selection and quickly eliminated from the populations (*Charlesworth, 2000*; *Uyenoyama and Newbigin, 2000*). A main unanswered question is whether functional specificities

remain stable over time or are subject to frequent turnover. Detailed theoretical analysis of the model of *Gervais et al. (2011)* showed that under a large portion of the parameter space, the introduced self-compatible intermediate is predicted to exclude its functional ancestor from the population. Secondary introduction of the compensatory mutation then effectively results in turnover of recognition specificities along allelic lines rather than in diversification per se. A turnover of recognition specificities may therefore be expected along each allelic lineage, rather than their long-term maintenance over evolutionary times. A similar process could also be occurring according to the progressive reinforcement model of *Chookajorn et al. (2004)*. Overall, while the large allelic diversity at the S-locus is one of the most striking features of this system, the evolutionary scenario and molecular route by which new S-alleles arise remain largely unknown. An essential limitation in the field is the crucial absence of direct experimental approaches.

## Results

To conclusively evaluate these models we used ancestral sequence reconstruction of S-haplotypes that are currently segregating in the plant *Arabidopsis halleri*. Ancestral sequence reconstruction is a new approach to 'resurrect' ancestral biological systems, whereby the sequence of an ancestral gene or allele is determined through phylogenetic analyses, then synthesized de novo and its functional properties (in terms of *e.g.* biochemical activity, conformation or binding capacity) are compared to those of its contemporary descendants. Although most S-haplotypes in *A. halleri* are typically so highly diverged that even sequence alignment can be challenging, previous work identified a pair of S-haplotypes (S03 and S28) with high phylogenetic proximity based on a fragment of the *SRK* gene (*Castric et al., 2008*). We took this opportunity to reconstruct, resurrect and phenotypically characterize their putative last common ancestral receptor SRKa ('a' for ancestor, *Figure 1*) into *A. thaliana*, which has previously been established as a model plant for mechanistic studies of SI (*Nasrallah et al., 2002*; *Tsuchimatsu et al., 2010*).

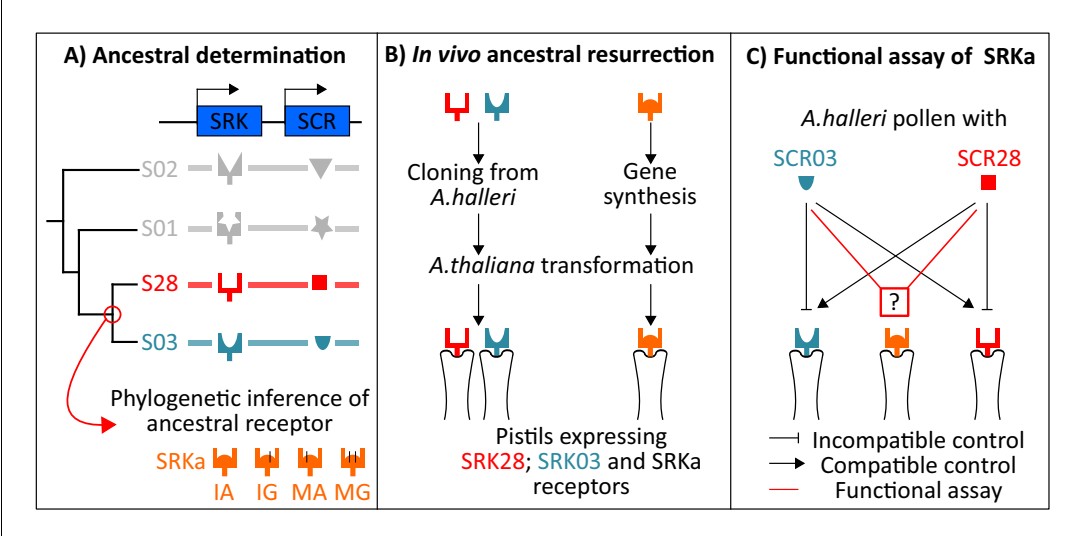

**Figure 1.** Experimental approach for the ancestral resurrection experiment. (**A**) The sequence of the putative last common ancestor of SRK03 and SRK28 was inferred by a phylogenetic approach using codon-based models implemented in PAML. Four different versions of SRKa were defined due to inference uncertainty at two aa positions. (**B**) SRK03 and SRK28 sequences were cloned from A. halleri DNA BAC clones, whereas SRKa sequences were obtained by gene synthesis. (**C**) Representation of the controlled cross program to decipher the specificity of SRKa.

The online version of this article includes the following figure supplement(s) for figure 1:

**Figure supplement 1.** Three models for the emergence of new self-incompatibility specificities.
**Figure supplement 2.** Maximum likelihood phylogenetic tree based on the 17 SRK alleles used for SRKa construction.
**Figure supplement 3.** Schematic representation of the molecular constructs used for *A. thaliana* transformation.

## Expression of S03 and S28 recognition specificities in *A. thaliana* by genetic transformation

Because of their relatively high sequence similarity, we first ensured that the S03 and S28 recognition specificities are indeed phenotypically distinct when expressed in *A. thaliana*. It was previously shown that the self-compatible model plant *A. thaliana* can mount a functional self-incompatibility response upon introduction of the S-locus genes *SCR* and *SRK* (*Boggs et al., 2009*; *Durand et al., 2014*; *Nasrallah et al., 2002*) but it is unclear whether transfer is possible for all *SCR* and *SRK* variants. We first assessed the activity of *AhSRK03* and *AhSRK28* promoters in transformed *A. thaliana* lines (*Figure 2—figure supplement 1*), which enabled us to identify the temporal window of *SRK* expression (*Figure 2—figure supplement 2*). Then, we used *A. halleri* pollen carrying either the S03 or S28 male determinant to validate that a proper SI response was indeed successfully transferred in transformed *A. thaliana* lines for each of our *SRK* receptors, and that this response was strictly specific toward either of their cognate ligand (two replicate transgenic lines for SRK03 and SRK28 each, *Figure 2*).

Abundant germination of pollen tubes occurred in crosses involving non-cognate SRK and SCR proteins, whereas crosses involving proteins from the same S-haplotype induced inhibition of pollen germination, which is characteristic of an incompatible response. Intensity of the incompatible response was then quantified by counting the number of germinated pollen grains at the pistil surface of two selected lines for each transgene (*Figure 2A and B*). Incompatible responses are characterized by less than 10 germinated pollen tubes whereas compatible crosses induced significant pollen germination, with more than 50 pollen tubes per stigma. This shows that cross-species pollinations of *A. thaliana* with *A. halleri* pollen can induce robust SI reactions, and also that S03 and S28 have distinct recognition specificities, despite their close phylogenetic proximity.

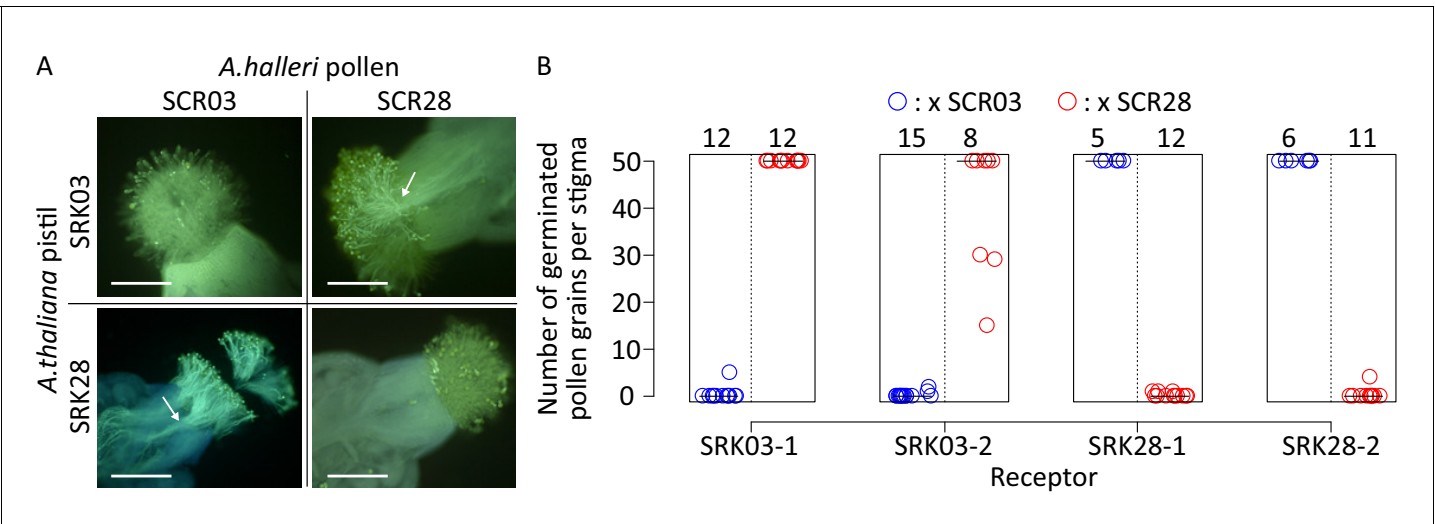

**Figure 2.** SRK03 and SRK28 are successfully expressed in *A. thaliana* and represent phenotypically distinct female recognition specificities. (**A**) Fluorescence microscopic observation of pistils from SRK-transformed *A. thaliana* plants pollinated with S03 and S28 pollen from *A. halleri*. A robust self-incompatibility reaction is observed between cognate alleles, whereas a compatible reaction is observed between non-cognate alleles. White arrows indicate pollen tubes that germinated in the stigma, and are specific to compatible reactions. Bar = 0.3 mm. (**B**) Number of germinated pollen grains per stigma after pollination. Two SRK03 and SRK28 lines were pollinated with *A. halleri* pollen expressing either S03 (blue) or S28 (red) specificities. The number of pollinated pistils for each pollination assay is indicated on the top of the figure. The median value for each cross is represented by a horizontal bar.

The online version of this article includes the following source data and figure supplement(s) for figure 2:

**Source data 1.** Number of germinated pollen grains per stigma after pollination.
**Figure supplement 1.** Localization of SRK expression by GFP florescence microscopy.
**Figure supplement 2.** Pattern of proSRK03 activity along development stages of floral buds.

## Phylogeny-based ancestral SRK reconstruction

We then used a phylogenetic approach to reconstruct the sequence of the most recent common ancestor of *SRK03* and *SRK28* (*Figure 3—figure supplement 1*). Based on a set of closely related *SRK* sequences from Arabidopsis and Capsella, we used the best fitting codon-based models implemented in PAML (*Yang, 2007*) to reconstruct the most likely ancestral amino acid sequence (SRKa). This most likely ancestral amino acid sequence had an Isoleucine (I) at position 208 and an Alanine (A) at position 305 and is therefore noted SRKa$^{IA}$. For the vast majority of the 56 amino acid residues where *SRK03* and *SRK28* differ (*Figure 3—figure supplement 1*), the ancestral reconstruction had high posterior probability (>0.95), but was more uncertain at four positions (*Figure 3—figure supplement 2*). Two of these sites were close (208) or within (305) hyper variable regions known to be functionally important (*Figure 3—figure supplement 1*; *Kusaba et al., 1997*). To take uncertainty into account and evaluate the impact of variation at these two individual sites, three additional ancestral sequences were generated and tested: SRKa$^{IG}$, SRKa$^{MA}$ and SRKa$^{MG}$. To avoid context effects, each of these four versions of *SRKa* was then surrounded by a promoter and a kinase sequence corresponding either to the native SRK03 (p03_SRKa_k03) or the native SRK28 (p28_SRKa_k28) variant. For each of these constructs, we obtained several replicate transgenic lines, resulting in a total of 28 different fixed homozygote single-copy transformed *Arabidopsis* lines (8 *SRKa$^{IA}$* lines; 7 *SRKa$^{IG}$* lines; 7 *SRKa$^{MA}$* lines and 6 *SRKa$^{MG}$* lines). Five lines showed very low levels of *SRKa* transcripts in pistils (*Figure 3—figure supplement 3*; possibly due to positional effects as is common in transgenic constructs) and were thus excluded from further analysis, leaving 23 transgenic lines for testing.

## Ligand specificity of the ancestral SRK

We then evaluated the recognition phenotype conferred by these putative ancestral SRK sequences by performing controlled pollination assays for all remaining 23 *SRKa* lines with pollen from *A. halleri* S03, *A. halleri* S28 and *A. thaliana* C24 (used as a control to validate receptivity of the acceptor pistil). The SI reaction was scored as the number of germinated pollen grains per stigma. All 23 tested SRKa expressing lines showed intense pollen germination with C24 pollen, idid not impair pistil fertility or ability to receive pollen (*Figure 3* and *Figure 3—figure supplement 4*).

With a single exception (line p03_SRKa$^{MA}$_k03_2), S28 pollen induced no consistent SI rejection in any of the lines expressing the different versions of the ancestral receptor, and was indistinguishable from the compatible control (C24, *Figure 3*). In stark contrast, SCR03 pollen induced robust incompatibility reactions in the vast majority (12 of 14) of the lines with amino acid I at position 208 (*SRKa$^{IA}$* and *SRKa$^{IG}$*, *Figure 3*). Hence, we conclude that the S03 haplotype has retained the ancestral recognition specificity largely unaltered, while the S28 haplotype underwent substantial modification and represents a novel recognition specificity, leading to functional diversification.

The incompatibility reactions involving our different ancestral SRK S-domain sequences were largely independent of the sequence context into which we introduced them (*Figure 3*, *Figure 1—figure supplement 3*). Similarly, the amino acid present at position 305 of the reconstructed sequence had no effect on the response: lines with either A or G produced rejection reactions equally. In contrast, only four of the lines with amino acid M at position 208 (SRKa$^{MA}$ and SRKa$^{MG}$) displayed incompatibility reactions, and these were weak (*Figure 3*). Hence, the single change, from the ancestral Isoleucine amino acid (I) at position 208 to Methionine (M), seems to be sufficient to almost fully inactivate the ancestral receptor's ability to recognise SCR03.

## Structural determinants of SRK specificity

We next sought to identify the structural determinants of the S03 and S28 specificities. We used the structural model of the interaction complex formed by SCR and SRK (*Ma et al., 2016*) to compare cognate and non cognate complexes in terms of number and nature of atomic contacts and predict how the ancestral receptor is expected to interact with either SCR03 or SCR28. As previously demonstrated by *Ma et al. (2016)*, SCR binding induces homodimerization of the extracellular domain of SRK (eSRK, *Figure 3—figure supplement 1*), forming eSRK:SCR heterotetramers composed of two SCR molecules bound to two SRK molecules (*Ma et al., 2016*). The switch in recognition specificity occurred along the branch leading to *SRK28*, and therefore must involve some of the 29 amino acid differences along this branch. In contrast, the 31 amino acid differences that accumulated along the

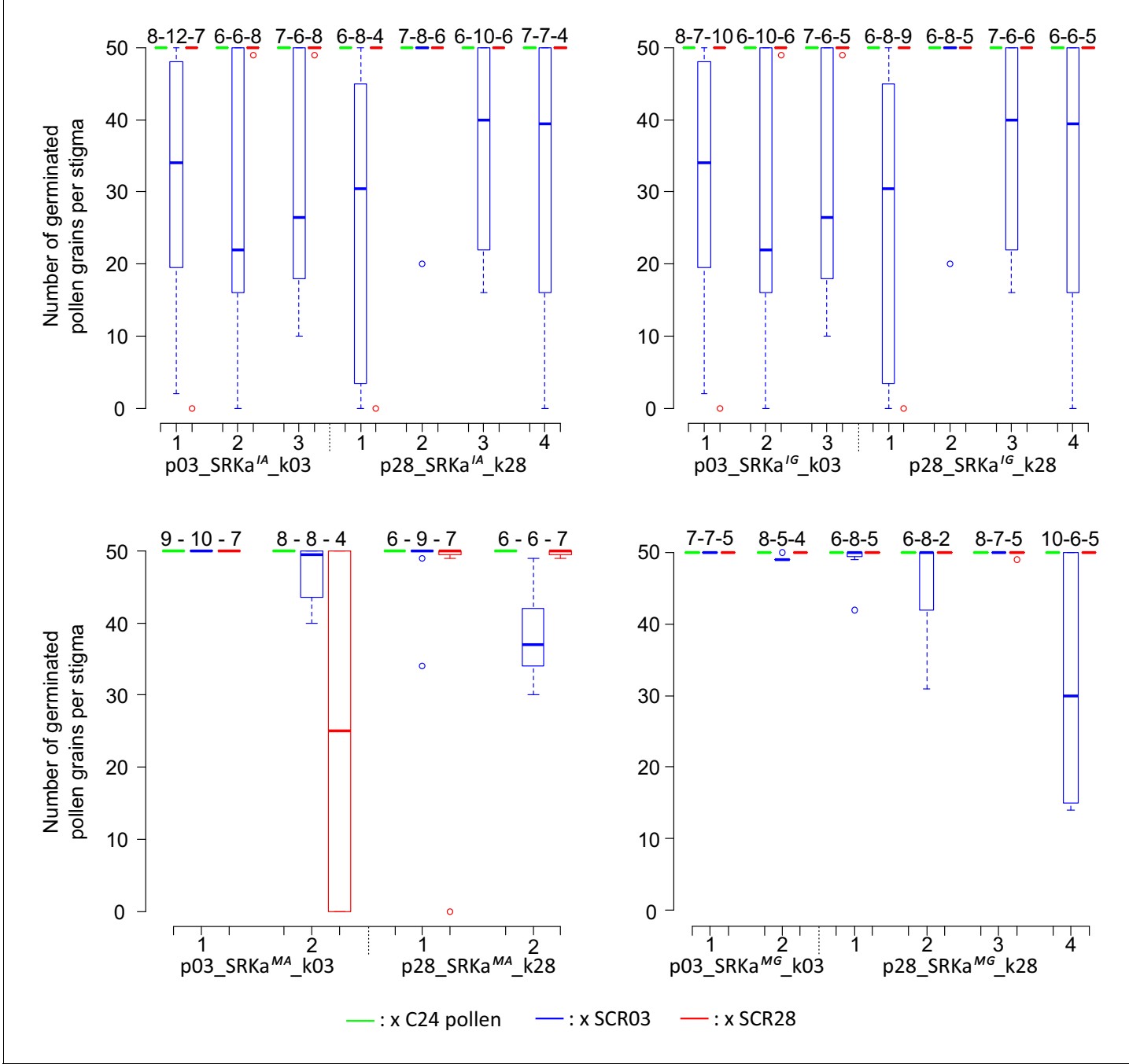

**Figure 3.** Incompatibility response of SRKa lines. Each SRKa line was pollinated with *A. thaliana* C24 (green), A. halleri S03 (blue) and S28 (red) pollen. The number of replicate pollinated pistils is indicated above each boxplot. The horizontal bar represents the median, the box delimitates the 25% and 75% percentiles, the bottom and top whiskers both represent 25% extreme percentiles and outliers are represented by individual dots.

The online version of this article includes the following source data and figure supplement(s) for figure 3:

**Source data 1.** Number of germinated pollen grains per stigma after pollination.

**Figure supplement 1.** Comparison between the S-domain of AhSRK03 and AhSRK28 and their common ancestor.

**Figure supplement 2.** Histogram presenting the distribution of amino acid (aa) probability values obtained with the best fitting-model M3FMutSel.

**Figure supplement 3.** Transgene expression analysis of the different SRK-transformed lines.

**Figure supplement 4.** Quantification of incompatibility response of *A. thaliana* lines with SRKa transcript levels below detection threshold.

**Figure supplement 4—source data 1.** Number of germinated pollen grains per stigma after pollination of *A. thaliana* lines with SRKa transcript levels below detection threshold.

branch leading to *SRK03* have not fully altered the recognition specificity. Overall, we find a slightly higher number of atomic contacts between the SCR and SRK molecules in the SRK03-SCR03 cognate complex (1543 contacts over 91 and 62 distinct SRK and SCR amino acids respectively) than in the SRK28-SCR28 cognate complex (1395 contacts over 89 and 57 distinct SRK and SCR amino acids respectively, *Supplementary file 1*). As already noted by *Ma et al. (2016)* in Brassica, and in line with the distribution of positively selected sites along the sequence (*Castric and Vekemans, 2007*), in both cases the interaction interface was mostly concentrated around the three 'hypervariable' portions of the SRK protein (*Figure 4*).

Contrary to our expectation, however, the non-cognate complexes did not differ substantially from the cognate complexes on the basis of their total number of atomic contacts (*Figure 4*). Specifically, the SRK03/SCR28 and SRK28/SCR03 non-cognate complexes were predicted to establish a total of 1546 and 1328 contacts, respectively, between atoms of their SCR and SRK molecules, very close to the 1543 and 1395 predicted contacts for the SRK03/SCR03 and SRK28/SCR28 cognate complexes (*Supplementary file 1*). They were also not characterized by any obvious steric clashes that would prevent the proper docking of SCR and SRK. Hence, the overall stability of the complex does not seem to be the primary determinant of recognition specificity. Consequently, the specificity of the interaction must be a function of some qualitative features rather than of the quantitative strength of the interaction.

While the majority of amino acids retained unchanged atomic contacts in cognate vs. non-cognate interactions, using a 5% threshold we identified six amino acid positions of the eSRK protein that displayed important differences in terms of contacts they establish with SCR in the cognate *vs.* the non-cognate complexes (*Figure 5—figure supplement 1B, C, E and F*). Four of these amino acid positions also displayed very contrasted interactions in the S03 vs. the S28 complexes (residues S201, R220, R279, W296, *Figure 5—figure supplement 1A and D*). For instance, the R residue at position 279 of SRK establishes three times more contacts with SCR in the SRK03/SCR03 than in the SRK28/SCR28 complex (*Figure 5*, *Figure 5—figure supplement 1A*). These four residues are therefore prime candidates for the determination of binding specificity. Intriguingly, three of the six amino acids were identical between SRK03 and SRK28, and yet interacted in sharply different ways with their respective SCR ligands. For instance, while identical between the SRK03 and SRK28 sequences, residues W296 and R279 established three to four times more contacts in the S03 than in the S28 complex, and the R220 residue is involved in twice as many contacts in the S28 than in the S03

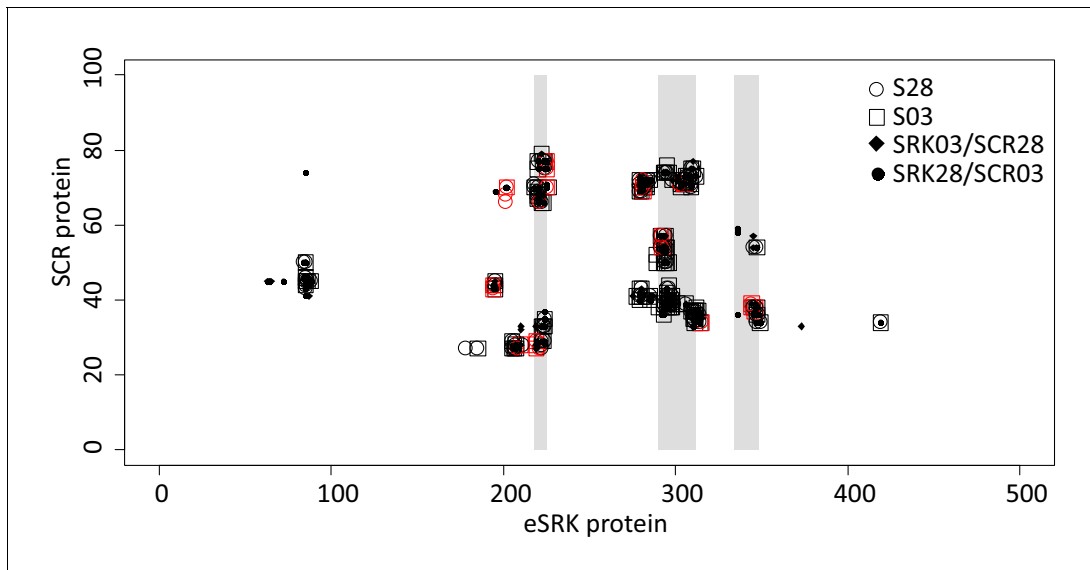

**Figure 4.** Map of atomic contacts from structural modelling of S03 and S28 complexes. The X and Y axis represent the eSRK and SCR protein respectively. Open circles and squares represent amino acid contacts between eSRK and SCR proteins in cognate S03 and S28 complexes respectively. Red symbols represent SRK amino acids that differ between SRK03 and SRK28. Full diamonds and dots correspond to amino acid contacts in non-cognate SRK03/SCR28 and SRK28/SCR03 complexes respectively. Hypervariable regions 1, 2 and 3 of eSRK are represented by vertical grey bars at position 219–225, 290–312 and 334–348 respectively (*Ma et al., 2016*).

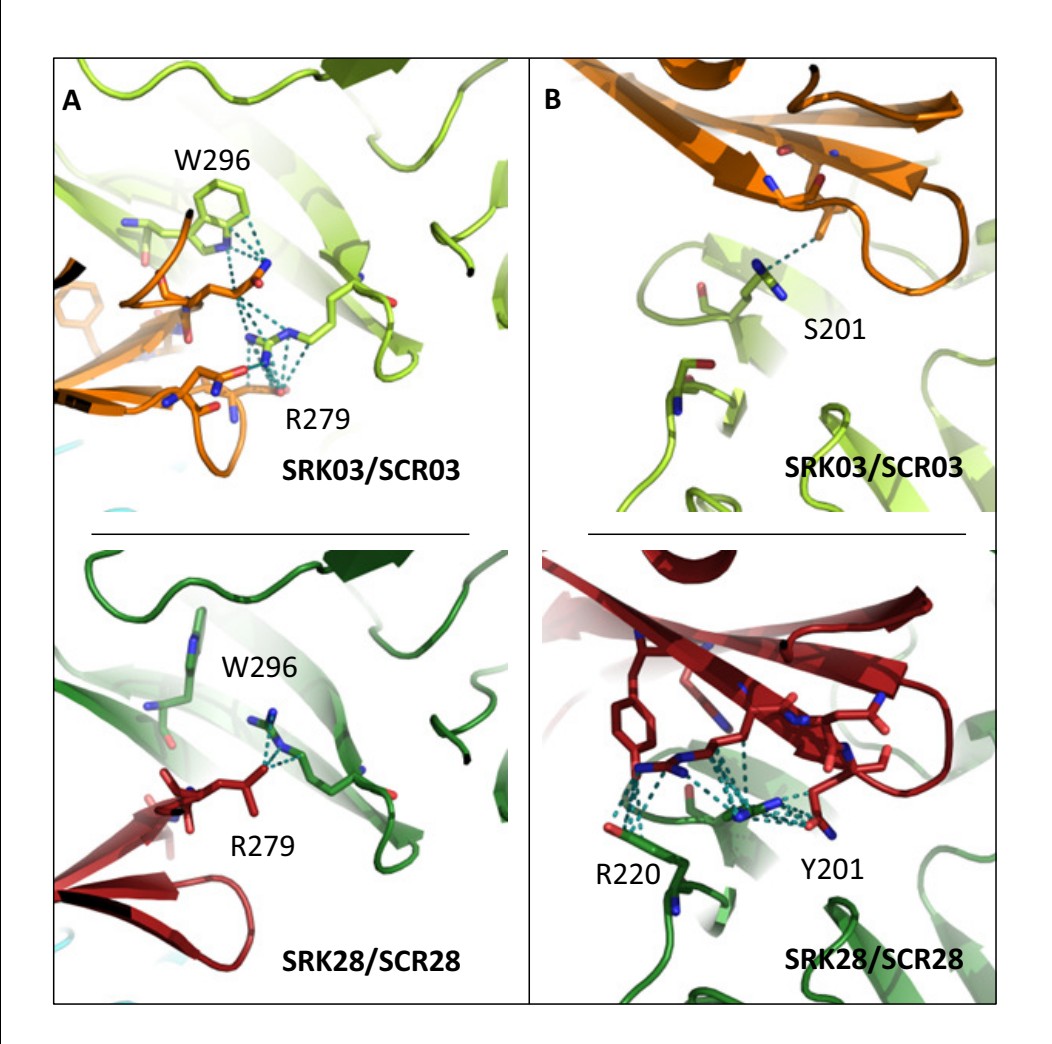

**Figure 5.** Four amino acid residues establish contrasted patterns of atomic contacts between SCR and SRK in the two cognate complexes. In (A), residues R279 and W296 of eSRK establish a large number of atomic contacts with SCR in the S03 complex (upper panel), but a very low number of contacts in the S28 complex (lower panel). The situation is reversed in (B), where residues S201 and R220 of eSRK establish a large number of atomic contacts with SCR in the S28 complex (lower panel), but a very low number of contacts in the S03 complex (upper panel). SRK chains are coloured in light and dark green for SRK03 and SRK28, respectively; SCR chains are coloured in orange and red for SCR03 and SCR28, respectively. Amino acid residues are shown in stick representation, with dotted lines indicating atom pair contacts below 4 Å, excluding hydrogen atoms. Note that for clarity a more stringent threshold was used to define atomic contacts here (4 Å) than in *Figure 5—figure supplement 1* and *Figure 6—figure supplement 1* (where a 5 Å threshold was used for a more comprehensive analysis), but the results are qualitatively similar.

The online version of this article includes the following figure supplement(s) for figure 5:

**Figure supplement 1.** Comparison of amino acid contacts between the different SCR/SRK complexes points to amino acids of SRK potentially involved in specificity of ligand recognition.

complex (*Figure 5*, *Figure 5—figure supplement 1A*). This suggests that involvement of these residues in the activity of the complex is mediated by substitutions at other positions along the protein sequence, for instance by displacing them spatially (*Figure 5*).

We then compared the predicted binding features of the functional form of the putative ancestral receptor (SRKa$^{IA}$) with both SRK03 and SRK28. When complexed with SCR03, 46 SRKa$^{IA}$ interacting pairs of amino acids were identical with SRK03, whereas only 24 were identical with SRK28. Similarly,

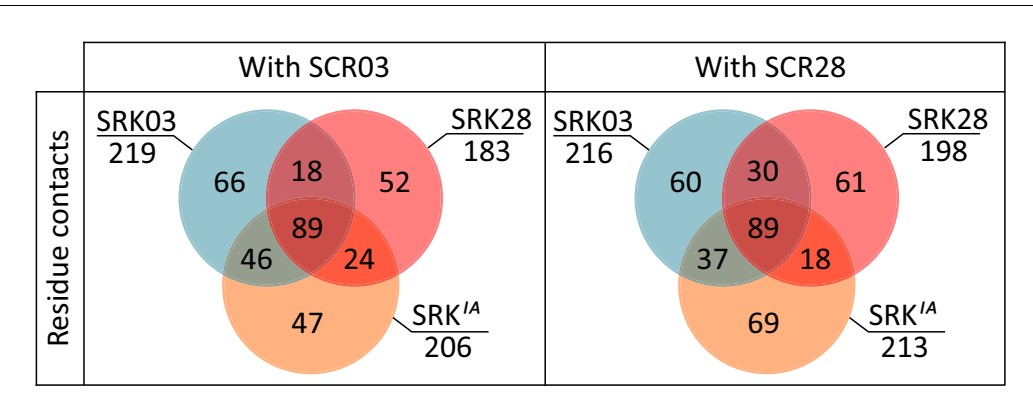

**Figure 6.** Comparison of predicted binding properties of SRK03, SRK28 and SRKa[IA] receptors when forming a complex with either the SCR03 or the SCR28 ligand. Residues in contact between SRKa[IA] and SCR are more often also in contact between SRK03 and SCR than between SRK28 and SCR. The Venn diagram indicates the number of contacts between amino acid residues that are shared or specific across variants of the SRK receptor and SCR. The online version of this article includes the following figure supplement(s) for figure 6:

**Figure supplement 1.** Venn diagram representations of binding features for SRK03, SRK28 and SRKa[IA] receptors coupled with either SCR03 or SCR28 ligand.

when complexed with SCR28, 37 SRKa[IA] interacting pairs of amino acid residues were identical with SRK03, but only 18 were identical with SRK28 (*Figure 6*). This trend is also visible at the level of individual atomic contacts and at the level of involved amino acids both in SRK and SCR molecules (*Figure 6—figure supplement 1*). Hence, even though there is no quantitative difference in how strongly SRK03 and SRK28 bind SCR03 and SCR28 (*Supplementary file 1*), the amino acids that come into contact in the two complexes are not identical, and the contacts established by SRKa[IA] with SCR qualitatively resemble more those established by SRK03 than those established by SRK28.

Overall, asymmetrical diversification of the ancestor into the phenotypically distinct S03 and S28 specificities is therefore associated with structural changes of the SRK receptor and the way it interacts with its SCR ligand. Despite similar levels of overall sequence divergence between SRKa and either SRK03 or SRK28, the binding properties of SRK03 have remained similar to those of SRKa[IA], while the way SRK28 is predicted to interact with the SCR ligand has diverged from its ancestral state, consistent with the phenotypic switch that has occurred along this lineage.

## Discussion

### Ancestral resurrection in a higher organism

Ancestral resurrection approaches have been used to decipher evolutionary processes in a number of biological systems including enzymes (*Starr et al., 2017*; *Tufts et al., 2015*), visual pigments (*Chang et al., 2002*) or transcription factors (*Pougach et al., 2014*). However, an essential limitation in these studies is that the ancestral protein function is typically determined in vitro, strongly limiting biological realism of the reconstructed function. To our knowledge, a single study involved a comparable in vivo ancestral resurrection approach in a higher organism, in the context of functional evolution of the alcohol dehydrogenase protein (ADH) in *Drosophila* lineages (*Siddiq et al., 2017*). Such whole organism studies are biologically most relevant, as they allow a focus on integrated phenotypes (here the self-incompatibility reaction, which has a strong and direct link with organismal fitness) rather than on highly reductionist phenotypes (such as in vitro assays of protein activity).

### Asymmetrical structural and functional divergence

Functional diversification of receptor-ligand interactions is a common process that takes place over long evolutionary times. However, in most cases we remain far from a predictive understanding of how the specific evolutionary forces imposed on a given interacting pair of proteins at the molecular

level translate into a given process of functional diversification. As a result, little is known about how diversification happens in nature, both in terms of whether some evolutionary pathways are preferentially followed over others and in terms of the molecular modifications involved. Here, we show that diversification of two S-alleles from their putative common ancestor proceeded through asymmetrical functional divergence, with one allele having largely retained its ancestral specificity and the other one having acquired a novel recognition specificity that was previously absent.

Overall, our results are thus compatible with the model of SI specificity diversification by compensatory mutation of *Gervais et al. (2011)* and one version (i.e. with maintenance of the ancestral specificity) of the progressive reinforcement model of *Chookajorn et al. (2004)*. However, they do not allow to discriminate between these two models, that is to determine whether a transient loss of self-recognition is required (*Uyenoyama et al., 2001*; *Gervais et al., 2011*). In contrast, the model of promiscuous dual-specificity intermediates (*Matton et al., 1999*; *Figure 1—figure supplement 1C*) can be rejected, because *Uyenoyama and Newbigin (2000)* have shown that a dual-specificity haplotype can be maintained in the population only in the absence of the ancestral specificity, while we show here that the ancestral specificity (S03) remained present. Similarly, the progressive reinforcement model with replacement of the ancestral specificity by two functionally diverged specificities can be rejected (*Figure 1—figure supplement 1B*).

## Long-term maintenance of SI specificity rather than rapid turnover

Models of SI evolution suggested that turnover of recognition specificities may be common over evolutionary times (*Gervais et al., 2011*). The observation that the ancestral SRKa$^{IA}$ recognition specificity was stably maintained along the SRK03 lineage demonstrates that no turnover event occurred over a substantial period of time (i.e. since S03 and S28 started to diverge from one another, *ca.* a few million years ago). The long-term maintenance of S-haplotype specificities was previously suggested by the comparison of certain *A. halleri* and *A. lyrata* S-lineages with their *A. thaliana* orthologs. Specifically, some *A. thaliana* accessions with haplogroup A were able to reject *A. halleri* pollen expressing the cognate AhSCR04 specificity (*Tsuchimatsu et al., 2010*), despite about 6 million years of divergence (*Hohmann et al., 2015*). Similarly, an overexpressed version of *A. thaliana SCR* haplogroup C was also able to elicit a SI reaction on pistils expressing the cognate *A. lyrata* AlSRK36 specificity (*Dwyer et al., 2013*). However, beside the fact that these were transspecific comparisons, it is important to note that *A. thaliana* has become predominantly self-fertilising, and no longer has an intact SI system due to alterations of SI components (*Bechsgaard et al., 2006*; *Shimizu and Tsuchimatsu, 2015*), so our results provide the first direct test of the hypothesis that recognition specificities remain stable, rather than undergoing recurrent rapid turnover.

Despite the obvious conservation of recognition specificity between SRKa$^{IA}$ and SRK03, we note that the SI response elicited by the reconstructed SRKa was overall weaker and more variable across replicate lines than those elicited by the native SRK constructs. This partial response could be due to incorrect inference of the ancestral receptor sequence. Ancestral reconstruction is based on probabilistic inference, and our results show that a single amino acid modification (I vs M at position 208) can entirely compromise the SI response. It is therefore possible that the putative ancestral receptor had a partly diminished capacity to elicit the SI reaction because of imperfect reconstruction. Alternatively, this partial response could be due to parallel changes to the SCR ligand, followed by amino acid substitutions occurring between SRKa and SRK03 to fine-tune the SRK03/SCR03 interaction. Resurrection of the ancestral SCRa ligand will be needed to distinguish between these possibilities.

## Structural determinants of the shift in binding specificity of a receptor-ligand interaction

Structural modelling of the cognate vs non-cognate complexes suggested that specificity of the receptor/ligand interaction depends on qualitative features rather than quantitative strength of the interaction. In other words, a ligand is predicted to bind its non-cognate just as well as its cognate receptor, but it activates only the proper one by interacting with a different set of amino acid residues at the binding interface. This may point to a mechanism of allosteric activation, whereby changes in the set (rather than number) of amino acid contacts with the ligand lead to changes in the shape of the receptor. Allosteric activation is important in the context of protein-protein interactions (*Kang et al., 2015*) as well as of the binding of transcription factors to their DNA binding sites

(*Gronemeyer and Bourguet, 2009*) and may be a general feature of molecular interfaces (*Ma et al., 2010*). In our system, the allosteric activation model suggests that only a few amino acid residues determine recognition specificity. This can be tested by directly measuring binding affinities, following *Ma et al. (2016)*. It will be interesting to determine experimentally whether this conclusion extends to more diverged pairs of SCR-SRK alleles, whose interactions are more common in SI systems than those between highly similar alleles studied here, because this balanced polymorphism has been maintained over a very long evolutionary time.

## Materials and methods

### Phylogeny-based ancestral SRK resurrection

#### Sequence collection
In order to reconstruct the ancestral sequence of the *AhSRK28* and *AhSRK03* SRK alleles, we first collected 17 full or partial sequences of the *SRK* first exon belonging to the haplogroup B/2. Seven allele sequences belonged to *A. halleri* (*AhSRK03*, *AhSRK08*, *AhSRK09*, *AhSRK19*, *AhSRK23*, *AhSRK27*, *AhSRK28*), six to *A. lyrata* (*AlSRK06*, *AlSRK08*, *AlSRK14*, *AlSRK18*, *AlSRK29*, *AlSRK39*) and four to *Capsella grandiflora* (*CgrSRK1*, *CgrSRK4*, *CgrSRK5*, *CgrSRK6*). For the *AhSRK19*, *AhSRK23* and *AhSRK27* alleles the *SRK* exon1 partial sequence was supplemented by Sanger sequencing. We added three sequences from the *SRK* haplogroup A3/3 as outgroups (*AhSRK04*, *AhSRK10*, *AhSRK29*). Accession numbers for the sequences are reported in *Supplementary file 2*. After preliminary analyses and in order to obtain a well-supported phylogenetic tree we removed three fast evolving sequences: *CgrSRK6*, *AhSRK08* and *AlSRK29*.

#### Phylogenetic analysis
Sequences were aligned with MACSE v1.2 (*Ranwez et al., 2011*). Based on the 1338 base pairs alignment, SRK phylogenetic trees were built with both Maximum Likelihood (ML) (PHYML 3.0, *Guindon et al., 2010*) and Bayesian methods (MrBayes 3.2.4, *Ronquist and Huelsenbeck, 2003*). For the ML analysis the model used was chosen according to jModelTest 2.1.10 (*Darriba et al., 2012*) using the Bayesian Information Criterion (BIC) (TPM3uf+Γ). Node stability was estimated by 100 non-parametric bootstrap replicates (*Felsenstein, 1985*). For the Bayesian inference, a codon model with a GTR+Γ+I model of substitution was used; two runs of four Markov chains were calculated simultaneously for 400,000 generations with initially equal probabilities for all trees and a random starting tree. Trees were sampled every 10 generations, and the consensus tree with posterior probabilities was calculated after removal of a 25% burn-in period. The average standard deviation of split frequencies between the two independent runs was lower than 0.01. The resulting topology was used to infer *SRKa* (*Figure 1—figure supplement 2*).

#### Ancestral SRK reconstruction
The ancestral reconstruction was conducted with the codeml program of the PAML 4.8 package (*Yang, 2007*) under six different models: M0, M0-FMutSel0, M0-FMutSel, M3, M3-FMutSel0, M3-FMutSel. M0 refers to a one-ratio model, which assumes a single ω ($d_n/d_s$) across branches and sites (*Goldman and Yang, 1994*), whereas M3 allows ω to vary across sites according to a discrete distribution (with 2 or three categories depending on the models) (*Yang et al., 2000*). The mutation-selection models incorporate parameters for mutation bias, with (FMutSel) or without (FMutSel0) selection on synonymous rate (*Yang and Nielsen, 2008*). Models were compared using Likelihood Ratio Tests (LRT) and Akaike Information Criteria (AIC). The best fitting-model was found to be M3FMutSel (*Supplementary file 3*), therefore the ancestral amino acid sequence reconstructed with this model was used to generate transgenic lines (SRKa[IA]). The ancestral amino acids inferred with the three 'M3' models were identical except for three sites with relatively lower probability: 33 (A or V), 208 (I or M) and 305 (A or G) (*Figure 3—figure supplement 2*). Two of these sites were close (208) or within (305) hyper variable regions and therefore potentially functionally important. To take into account these uncertainties, three additional transgenic lines were generated with a different SRKa and tested: SRKa[IG], SRKa[MA] and SRKa[MG], in order to evaluate all four combinations of two amino acids at these two sites.

## Generation and selection of *A. thaliana* transgenic lines

We generated 12 series of *A. thaliana* C24 transgenic plants (*AhSRK03*; *AhSRK28*; *AhSRK03p:GFP*; *AhSRK28p:GFP*; *p03_SRKa$^{IA}$_k03*; *p28_SRKa$^{IA}$_k28*; *p03_SRKa$^{IG}$_k03*; *p28_SRKa$^{IG}$_k28*; *p03_SRKa$^{MA}$_k03*; *p28_SRKa$^{MA}$_k28*; *p03_SRKa$^{MG}$_k03* and *p28_SRKa$^{MG}$_28*). All DNA amplifications were performed with the primeSTAR DNA polymerase (Takara, Japan) and all constructs were validated by SANGER sequencing. We used gateway vectors (Life Technologies, USA) for expression of transgenes in *A. thaliana*. *AhSRK03, AhSRK28, proAhSRK03* and *proAhSRK28* were amplified using attB1 and attB2 containing primers (*Supplementary file 4*) from BAC sequences of *A. halleri* S03 and S28 (KJ772378-KJ772385 and KJ461475-KJ461478 respectively, *Goubet et al., 2012*). Amplification products were inserted by BP recombination into the pDONR 221 plasmid. Promoters pro*AhSRK03* and pro*AhSRK28* designed for p03_SRKa_k03 and p28_SRKa_k28 constructs respectively were amplified with attB4 and attB1r containing primers (*Supplementary file 4*). Amplification products were inserted by BP recombination into a pENTR-P4-P1R plasmid. Kinase domains of AhSRK03 and AhSRK28 were amplified with attB2r and AttB3 containing primers (*Supplementary file 4*). Amplification products were inserted by BP recombination into a pENTR-P2R-P3 plasmid.

Constructs expressing native SRK (*AhSRK03* and *AhSRK28*) and GFP expressing constructs (*AhSRK03p:GFP* and *AhSRK28p:GFP*) were generated by LR reaction between entry clones and the destination vectors pK7WG or pKGWFS7.0. SRKa constructs were generated by LR triple recombination into the pK7m34GW destination vector with i) attL4 and attR1 promoters entry clones, ii) synthesized putative ancestral S-domain surrounded by attL1 and attL2 sequences and iii) attR2 and attL3 kinase domain entry clones. Details of the molecular constructs are displayed in *Figure 1—figure supplement 3*. *A. thaliana* plants were grown under 16 hr/20 °C day, 8 hr/18°C night and 70% humidity greenhouse conditions. When plants displayed approximately 20 flowers, transformation was performed by *Agrobacterium tumefaciens*-mediated floral deep according to *Logemann et al. (2006)*. After seed harvesting, single insertion homozygous lines were selected *via* multiple rounds of antibiotic selection on selective medium as described in *Zhang et al. (2006)*.

## Microscopy

GFP expression in *AhSRK03p:GFP* and *AhSRK28p:GFP* and the results of cross pollination experiment were monitored under UV light using an Axio imager two microscope (Zeiss, Oberkochen, Germany) coupled with a HXP120 Light source (LEJ, Jena, Germany). Pictures were taken with an Axiocam 506 color camera (Zeiss, Oberkochen, Germany) and read with the Zeiss ZEN two core software package. For *GFP* expression, floral buds, pieces of leaves and floral hamp cross sections from *AhSRK03p:GFP* and *AhSRK28p:GFP* were observed. For pollination assays, the number of germinated pollen grains per stigma was counted after aniline blue staining up to 50 pollen tubes.

## Aniline blue staining

To evaluate the SI response of transgenic lines, floral buds of developmental stage 9–11 were emasculated at day-1. At day-2, they were pollinated with frozen *A. halleri* pollen expressing S03 or S28 specificity. Pistils were harvested 6 hr after pollination and fixed in FAA (4% formaldehyde, 4% acetic acid, ethanol) overnight. On day-4, pollinated stigmas were washed three times in water, incubated 30 min in NaOH 4M at room temperature, washed three times in water and conserved for 30 min in the last bath. Finally, pistils were incubated overnight in aniline blue (K$_3$PO$_4$0.15 M, 0.1% aniline blue) and mounted between microscopic slides at day-5.

## Expression analysis

RNA extraction was performed onto stages 11–14 floral buds with the nucleospin RNA Plus Extraction kit (Macherey Nagel, GmbH and Co. DE). For each line, floral buds were harvested onto three different individuals. cDNA synthesis was performed using RevertAid Reverse Transcriptase (Thermo Fisher Scientific, Massachusetts, USA). 1 µg of extracted RNA was mixed with 0.2 µg of Random hexamer primer, psq H$_2$O 12.5 µL. After incubation at 65°C, 4 µL of 5X reaction buffer; 0,5 µL of Riboblock RNase inhibitor (Thermo Fisher Scientific, Massachusetts, USA), 10 mM of each dNTP and 200 U of RevertAid Reverse Transcriptase were added and incubated firstly at 25°C and then 10 min at 42°C. cDNA quantification was performed in triplicate using LightCycler 480 Instrument II (Roche molecular systems, Inc, Pleasanton, USA). cDNA amplification of *SRK* fragments was performed with

forward primer located on the S-domain (For: AGGAATGTGAGGAGAGGTGC) and the reverse primer located on the second exon (Rev: TCCTACTGTTGTTGTTGCCC). According to *Liu et al. (2007)*, Ubiquitin was used as housekeeping gene (For: CTGAGCCGGACAGTCCTCTTAACTG; Rev: CGGCGAGGCGTGTATACATTTGTG).

### Interacting protein modelling

All structural models were created using MODELLER (*Sali and Blundell, 1993*; *Fiser et al., 2000*), applying a template-based docking approach with the eSRK9:SCR9 X-Ray structure (*Ma et al., 2016*) as reference. Input sequence alignments were created using multiple sequence alignments of all relevant SRK sequences and of all relevant SCR sequences. The resulting models were subsequently optimized using the GalaxyRefineComplex web service (*Heo et al., 2016*).

The top models for each receptor-ligand pair were then considered and the number of atomic contacts between chains was counted using a 5 Å atom-atom cut-off (e.g. *Lensink et al., 2017*). Hydrogen atoms were excluded from the calculation. Visualization was done, and for clarity images were made with a 4 Å atom-atom cut-off using the PyMol molecular graphics system, version 1.7.2.1, Schrodinger LLC.

## Acknowledgements

This research received support through a grant from the European Research Council (NOVEL project, grant #648321). The authors thank the French Ministère de l'Enseignement Supérieur et de la Recherche, the Hauts de France Region and the European Funds for Regional Economical Development for their financial support to this project. They also thank S Billiard, I Fobis-Loisy, E Durand, D Charlesworth, M Uyenoyama and J Nasrallah for useful discussions.

## Additional information

### Funding

| Funder | Grant reference number | Author |
|---|---|---|
| H2020 European Research Council | Novel project grant #648321 | Vincent Castric |

The funders had no role in study design, data collection and interpretation, or the decision to submit the work for publication.

### Author contributions

Maxime Chantreau, Data curation, Formal analysis, Investigation, Methodology; Céline Poux, Data curation, Software, Formal analysis, Investigation, Methodology; Marc F Lensink, Guillaume Brysbaert, Software, Formal analysis, Investigation, Methodology; Xavier Vekemans, Supervision; Vincent Castric, Conceptualization, Resources, Supervision, Funding acquisition, Validation, Investigation, Project administration

### Author ORCIDs

Maxime Chantreau (iD) http://orcid.org/0000-0002-2844-1989
Céline Poux (iD) https://orcid.org/0000-0002-9379-2769
Guillaume Brysbaert (iD) https://orcid.org/0000-0002-6807-6621
Vincent Castric (iD) https://orcid.org/0000-0002-4461-4915

### Decision letter and Author response

Decision letter https://doi.org/10.7554/eLife.50253.sa1
Author response https://doi.org/10.7554/eLife.50253.sa2

## Additional files

### Supplementary files

• Supplementary file 1. Detail of the different SRK/SCR interaction protein modelling. Following the eSRK:SCR template structure where two SCR molecules interact with two SRK proteins to form a heterotetramer (*Ma et al., 2016*), we indicate the two SRK molecules with their chain identifier A and B and the two SCR molecules with G and H. For each complex, the number of amino acids involved and the number of atomic contacts are defined for each protein chain interaction (AG, AH, BG and BH). Underlined numbers in the column 'involved aa' correspond to the number of amino acids involved in both cognate and non-cognate interactions.

• Supplementary file 2. Accession numbers for the sequences used in the phylogenetic reconstruction.

• Supplementary file 3. PAML ancestral analyses of the SRK protein and model comparison. np is the number of parameters in the model; lnL is the log likelihood score; AIC (Akaike Information criterion = $-2*lnL+2*np$) is a measure of the goodness of fit of an estimated statistical model; ω is the nonsynonymous/synonymous substitution ratio; LR is the likelihood ratio: df is the degree of freedom in LRT (Likelihood Ratio Test); *** Highly significant (p-value<0.0001).

• Supplementary file 4. Gateway primers used for molecular constructs.

• Supplementary file 5. Key resources Table.

• Transparent reporting form

### Data availability

All data generated or analysed during this study are included in the manuscript and supporting files.

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
