## [Decision Letter]

**Acceptance summary:**

Self-incompatibility in the Brassicaceae is mediated by the S locus, which encodes two interacting proteins, a female-expressed S-locus Receptor Kinase and a male-expressed ligand, S-Locus Cysteine-rich Protein (SCR). Pollen is rejected when SRK and SCR are encoded by the same haplotype. Highly diverged allelic variants segregate in natural populations, but how this allelic diversity arose from an ancestral haplotype is unresolved. The authors use ancestral reconstruction to investigate how different S haplotypes might have arisen – which is interesting in itself but also as a model for the general question of how genes diversify despite constraints due to specific interactions. A main strength of this work is that four plausible ancestral alleles are reconstructed (and replicated). They then tested 3 proposed models, by introducing various constructs of A. halleri SRK into *A. thaliana*. Their results provide a convincing answer: favoring compensatory mutations and SC intermediates, and rejecting 2 other proposed models.

**Decision letter after peer review:**

Thank you for submitting your article "Asymmetrical diversification of the receptor-ligand interaction controlling self-incompatibility in Arabidopsis" for consideration by *eLife*. Your article has been reviewed by three peer reviewers, including Sheila McCormick as the Reviewing Editor and Reviewer #1, and the evaluation has been overseen by a Reviewing Editor and Christian Hardtke as the Senior Editor. The following individual involved in review of your submission has agreed to reveal their identity: Nick H Barton (Reviewer #3).

The reviewers have discussed the reviews with one another and the Reviewing Editor has drafted this decision to help you prepare a revised submission.

Summary:

This impressive paper sets out to distinguish how different S haplotypes have arisen. The authors use ancestral reconstruction to investigate S locus evolution – which is interesting in itself but also, as the authors explain, a model for the general question of how genes diversify despite constraints due to specific interactions. A key issue in such studies is the reliability of the reconstruction. However, four plausible ancestral alleles are reconstructed (and replicated), which is a strength.

There are 3 ideas in the field, and to test them the authors introduce various constructs of A. halleri SRK into *A. thaliana*. Their results provide a convincing answer (i.e. they favor the one of compensatory mutations and SC intermediates (Gervais).

Essential revisions:

The experiments are easy to follow and the discussion is thoughtful, but could be shortened by removing unnecessary repeats. For instance, the first paragraph just repeats what has already been stated perfectly clearly. Discussion subsection “Asymmetrical structural and functional divergence” can also be shortened considerably, and these shortenings will help readers come to the most important questions that they will have in mind at this point – i.e., which of the models outlined in the Introduction are excluded. Given the value of the work, it is worthwhile to make the text as clear as possible.

This section needs slight revision, to something like the following: "Overall, our results thus unambiguously reject two previously proposed models of SI specificity diversification. The model of gradual divergence of pairwise SRK-SCR affinities along lineages (Chookajorn et al., 2004) is rejected because it predicts that both descendent alleles would have been functionally distinct from the ancestor. The model of promiscuous dual-specificity intermediates (Matton et al., 1999) is also rejected, because Uyenoyama and Newbigin, 2000, have shown that a dual-specificity haplotype can be maintained in the population only in the absence of the ancestral specificity, while we show here that the ancestral specificity (S03) remained present. In contrast, the remaining scenario (in which diversification proceeds through transient self-compatible intermediates; Gervais et al., 2011; Uyenoyama et al., 2001) is compatible with our observation of long-term maintenance of functional specificities".

Please consider stating putative ancestral throughout or "ancestral", as you only surmise that this could be the scenario. Similarly, the terms "resurrected" and "now extinct" are suppositions.

In later experiments they analyze potential amino acid contacts in the synthesized SRKs and SCRs. Using the number of contacts seems somewhat crude; are there other cases that can be cited where this sort of metric has succeeded?

Subsection “Asymmetrical structural and functional divergence”: – One cannot conclude that specificities evolve through cladogenetic change rather than anagenetic. The latter refers to the question of whether changes are associated with speciation events. Here, the observation is that function has been preserved along one lineage but not the other. Does this really address whether speciation is associated with functional divergence?

Discussion section: It may be that "the specificity of the receptor/ligand interaction depends on qualitative features rather than quantitative", but that seems more general than " allosteric activation", which refers to a change in specificity caused by a change in the shape of the protein. If the argument is that allostery is involved, it needs to be made more clearly.

Please give more information (i.e. add a figure) about the amino acid sequence differences, including the total number of differences between the SRK sequences used in the study, and their locations in relation to the recognized functional domains of the protein. Subsection “Phylogeny-based ancestral SRK reconstruction” mentions "the vast majority of amino acid residues where SRK03 and SRK28 differ", but doesn't tell us the actual number or any other information. Subsection “Ligand specificity of the ancestral SRK” says that the study examined the S-domain, but the domains have not been clearly outlined or explained. It would also have helped understanding to be told about the differences from the inferred ancestral protein sequence. Without these details, it is difficult to understand the naming of the sequences that were tested. A brief description of the phylogenetic approach used to reconstruct the sequence of SRK03 and SRK28's most recent common ancestor would also be helpful.

Can Figure 1A be revised a bit? The shapes are an over-simplification of the experiments, as there were 4 different versions tested, the figure implies only 1.

---

## [Author Response]

Essential revisions:The experiments are easy to follow and the discussion is thoughtful, but could be shortened by removing unnecessary repeats. For instance, the first paragraph just repeats what has already been stated perfectly clearly.

We have removed this first paragraph.

Discussion subsection “Asymmetrical structural and functional divergence” can also be shortened considerably, and these shortenings will help readers come to the most important questions that they will have in mind at this point – i.e., which of the models outlined in the Introduction are excluded. Given the value of the work, it is worthwhile to make the text as clear as possible.

We have shortened this paragraph by one half and now come more directly to the core questions.

This section needs slight revision, to something like the following: "Overall, our results thus unambiguously reject two previously proposed models of SI specificity diversification. The model of gradual divergence of pairwise SRK-SCR affinities along lineages (Chookajorn et al., 2004) is rejected because it predicts that both descendent alleles would have been functionally distinct from the ancestor. The model of promiscuous dual-specificity intermediates (Matton et al., 1999) is also rejected, because Uyenoyama and Newbigin, 2000, have shown that a dual-specificity haplotype can be maintained in the population only in the absence of the ancestral specificity, while we show here that the ancestral specificity (S03) remained present. In contrast, the remaining scenario (in which diversification proceeds through transient self-compatible intermediates; Gervais et al., 2011; Uyenoyama et al., 2001) is compatible with our observation of long-term maintenance of functional specificities".

We have revised this paragraph (subsection “Asymmetrical structural and functional divergence”) to better clarify which models can be rejected, which led us to expand a bit the paragraph where we present the models in the Introduction. In particular we placed a stronger focus on the persistence of the ancestral specificity after diversification, which is predicted by Gervais et al., 2011 and possibly also by Chookajorn et al., 2004. We note that functional turnover is another possible outcome for both models, but the lack of formal theoretical analysis of the latter prevents more precise distinction at this stage. We feel that this more precise presentation of the models will make the implications of our results more evident to the readers.

Please consider stating putative ancestral throughout or "ancestral", as you only surmise that this could be the scenario. Similarly, the terms "resurrected" and "now extinct" are suppositions.

We have changed the wording to “putative ancestral” at most places in the text.

As requested, we have changed the term “resurrected” to “reconstructed” in the Abstract. At other places in the text we still refer to a “resurrection” experiment, as this wording is now well established in the literature.

We have retained in the Abstract the mention to the fact that the putative ancestor is “now extinct”, as this is indeed reflecting the reality and is not a supposition.

In later experiments they analyze potential amino acid contacts in the synthesized SRKs and SCRs. Using the number of contacts seems somewhat crude; are there other cases that can be cited where this sort of metric has succeeded?

Computational predictions of protein-protein interaction use several quantities to estimate the quality of the predicted model, including the fraction of native contacts. This quantity is purely based on interatomic and/or inter-residue distances criteria, as we have used here and is common practice in the field. We have added a reference to the CAPRI experiment (Lensink et al., 2017) that evaluates the performance of different metrics used to evaluate the quality of protein docking (including the number of atomic contact).

Subsection “Asymmetrical structural and functional divergence”: One cannot conclude that specificities evolve through cladogenetic change rather than anagenetic. The latter refers to the question of whether changes are associated with speciation events. Here, the observation is that function has been preserved along one lineage but not the other. Does this really address whether speciation is associated with functional divergence?

We have removed this sentence.

Discussion section: It may be that "the specificity of the receptor/ligand interaction depends on qualitative features rather than quantitative", but that seems more general than " allosteric activation", which refers to a change in specificity caused by a change in the shape of the protein. If the argument is that allostery is involved, it needs to be made more clearly.

We have modified the sentence to better explain the allosteric activation model.

Please give more information (i.e. add a figure) about the amino acid sequence differences, including the total number of differences between the SRK sequences used in the study, and their locations in relation to the recognized functional domains of the protein. Subsection “Phylogeny-based ancestral SRK reconstruction” mentions "the vast majority of amino acid residues where SRK03 and SRK28 differ", but doesn't tell us the actual number or any other information. Subsection “Ligand specificity of the ancestral SRK” says that the study examined the S-domain, but the domains have not been clearly outlined or explained. It would also have helped understanding to be told about the differences from the inferred ancestral protein sequence.

We have modified Figure 3—figure supplement 1 to make this information more apparent (number of amino acid sequence differences between SRK03 and SRK28 and with SRKa, functional domain annotation of the SRK protein). We also report these numbers directly in the text.

Without these details, it is difficult to understand the naming of the sequences that were tested. A brief description of the phylogenetic approach used to reconstruct the sequence of SRK03 and SRK28's most recent common ancestor would also be helpful.

We now explain directly the naming of the sequences: “This most likely ancestral amino acid sequence had an Isoleucine (I) at position 208 and an Alanine (A) at position 305 and is noted SRKa^IA”^.

Can Figure 1A be revised a bit? The shapes are an over-simplification of the experiments, as there were 4 different versions tested, the figure implies only 1.

We have improved this figure to represent the four reconstructed ancestors.